# Effective screening of Coulomb repulsions in water accelerates reactions of like-charged compounds by orders of magnitude

Adam Kowalski [1,7], Krzysztof Bielec [2,7], Grzegorz Bubak [1,7], Pawel J. Żuk [1,3], Maciej Czajkowski[4], Volodymyr Sashuk [1]✉, Wilhelm T. S. Huck [5]✉, Jan M. Antosiewicz [6]✉ & Robert Holyst [1]✉

The reaction kinetics between like-charged compounds in water is extremely slow due to Coulomb repulsions. Here, we demonstrate that by screening these interactions and, in consequence, increasing the local concentration of reactants, we boost the reactions by many orders of magnitude. The reaction between negatively charged Coenzyme A molecules accelerates ~5 million-fold using cationic micelles. That is ~$10^4$ faster kinetics than in 0.5 M NaCl, although the salt is ~$10^6$ more concentrated. Rate enhancements are not limited to micelles, as evidenced by significant catalytic effects ($10^4$–$10^5$-fold) of other highly charged species such as oligomers and polymers. We generalize the observed phenomenon by analogously speeding up a non-covalent complex formation—DNA hybridization. A theoretical analysis shows that the acceleration is correlated to the catalysts' surface charge density in both experimental systems and enables predicting and controlling reaction rates of like-charged compounds with counter-charged species.

The occurrence of any chemical reaction requires its reactants to meet in the correct spatiotemporal manner and with sufficient energy. Whether a given transformation occurs is controlled by the nature of the charge; positive or negative, its quantity, surrounding atoms, and distribution among molecules. For the effective product formation between like-charged molecules, additional energy is required to overcome Coulomb repulsions. Thus, such reactions in pure water can take days or even weeks[1]. In biological systems, these reactions are usually accelerated by enzymes acting as catalysts. Warshel has shown that electrostatic effects and the stabilization of charges provide a key catalytic contribution[2]. There has been a vast interest in synthetic enzymes or enzyme-mimics such as nanozymes, but capturing the properties of the active site of enzymes with respect to charge stabilization has been challenging and typically highly selective for specific reactions[3–5].

Recently, we discovered a significant acceleration of the chemical reaction between negatively charged reactants when positively charged polymers were added, which we attributed to the sliding of the reactants along molecular tracks (so-called diffusive binding)[6]. Inspired by this finding, we aim to answer whether the rate enhancement by 'counter-charged' species could be a general phenomenon and explain the theoretical basis of such electrostatic catalysis.

In this work, we systematically study how the reaction kinetics of two independent experimental systems (i.e., covalent and non-covalent product formation) were influenced by the nature of the charges and the form in which charges were introduced, i.e., salt ions, charged monomers (or their oligomers), charged micelles, and charged polymers. As a first model reaction, we investigate the reaction between coenzyme A (CoA) and bromo-N-

[1]Institute of Physical Chemistry, Polish Academy of Sciences, Warsaw 01-224, Poland. [2]University of Zurich, Department of Chemistry, Zurich CH-8057, Switzerland. [3]Lancaster University, Department of Physics, Lancaster LA1 4YB, UK. [4]University of Oxford, Department of Chemistry, Oxford OX1 3TA, UK. [5]Institute for Molecules and Materials, Radboud University, Nijmegen 6525 AJ, Netherlands. [6]Institute of Experimental Physics, Biophysics Division, University of Warsaw, Warsaw 02-093, Poland. [7]These authors contributed equally: Adam Kowalski, Krzysztof Bielec, Grzegorz Bubak. ✉e-mail: vsashuk@ichf.edu.pl; w.huck@science.ru.nl; jantosi@fuw.edu.pl; rholyst@ichf.edu.pl

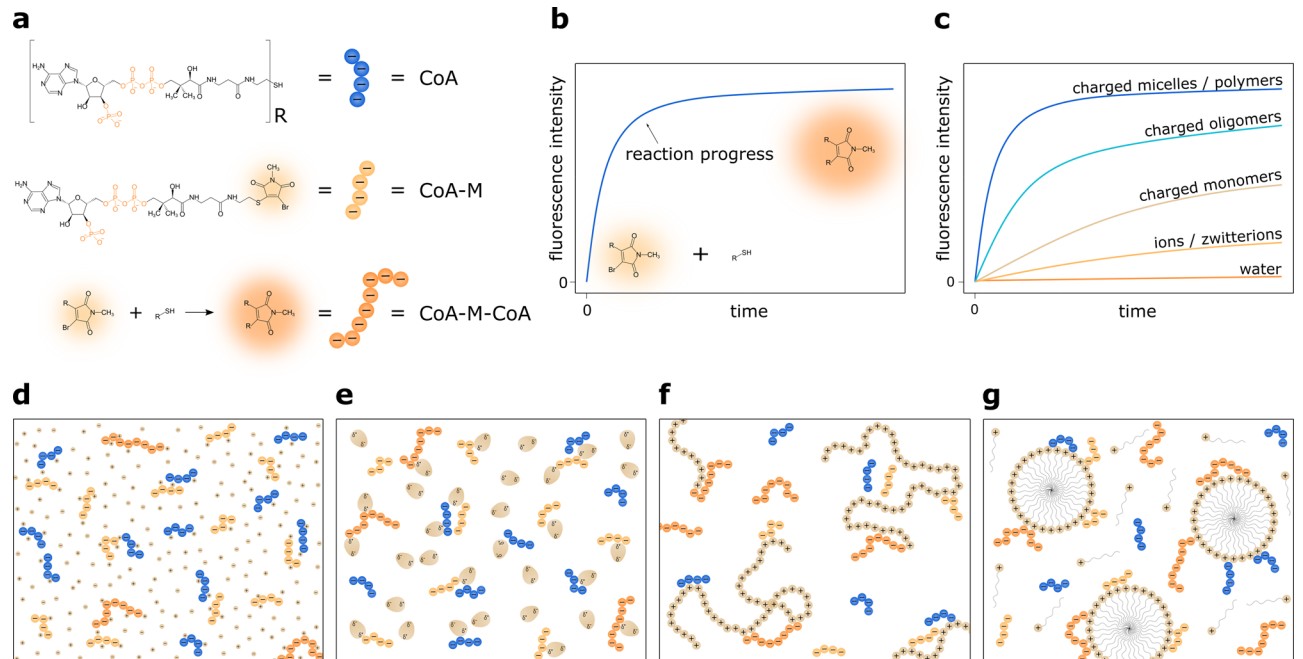

**Fig. 1 | Schematic representation of the experimental system. a** The reaction between negatively charged coenzymes A (CoA) is used as a model reaction. One of them−CoA-M−has fluorescent moiety. The reaction between reactants occurs through the substitution of bromine in the CoA-M by the thiol group of CoA. This process is practically irreversible, and as a product, CoA-M-CoA molecule is formed. **b** The product formation is accompanied by an increase in fluorescence intensity. **c** Here, we show how the charge screening and enhancement of local concentration of reactants at the oppositely charged molecules' surfaces increase reaction rates of Coenzyme A molecules. We investigate the reaction speed up in the aqueous solution in the presence of **d** ions and charged monomers, **e** zwitterions, **f** charged oligomers and polymers, and **g** charged micelles.

methylmaleimide substituted coenzyme A (CoA-M), as shown in Fig. 1, which can be followed conveniently by the increase in fluorescence intensity. By introducing positively charged micelles to the reaction system, an up to 5 million-fold rate enhancement is observed compared to the same reaction in water. In contrast to common micellar catalysis, the reactions occur exclusively on the surface of the charged micelles[7]. Apart from the reaction acceleration for positively-charged micelles of cetrimonium chloride (CTAC), benzethonium chloride (BTC), and cetylpyridinium chloride (CPC), we prove that when anionic, (sodium dodecyl sulfate - SDS) or neutral (Brij L23) surfactants are used, the reaction rates are similar to the one in water or buffer. Moreover, different forms of cations result in different reaction rates allowing us to tune the reaction time scales from seconds to hours. As a second experimental system, we analyze non-covalent complex formation between complementary DNA strands, as these reactants also strongly repel each other in an aqueous environment without the presence of screening ions. Similar to the first reaction system, the DNA duplex formation rate increases by orders of magnitudes (up to three) in solutions containing cationic micelles compared to 1 mM phosphate buffer, demonstrating the observed phenomenon's universality. Based on the above examples, we theoretically explain the observed experimental phenomenon and demonstrate a quantitative model predicting the acceleration rates.

## Results and discussion

The reaction between CoA and CoA-M is an irreversible second-order process that can be written as the reaction between reactants $A$ and $B$ to form product $AB$:

$$A + B \xrightarrow{k} AB, \tag{1}$$

where $k$ is the reaction rate constant. The kinetic equation for this process in relation to the product concentration change over time,

$[AB]_t$, takes the form

$$\frac{1}{[AB]_f - [AB]_t} = k_t \cdot t + \frac{1}{[AB]_f}, \tag{2}$$

where $t$ is time and $[AB]_f$ is the final product concentration. During CoA-M-CoA formation, the fluorescence intensity increases, which is directly proportional to its increase in concentration. Thus, the change in product concentration over time can be written as the change in fluorescence intensity over time, $I_t$, and the final product concentration as the fluorescence intensity at the end of the reaction, $I_f$:

$$\frac{1}{I_f - I_t} = k_t \cdot t + \frac{1}{I_f}. \tag{3}$$

By introducing a catalyst into the system, intermediates may occur during the formation of the product (the problem of intermediates is explained in more detail in Supplementary Note 3). However, they react reversibly with the reactants and do not significantly affect the kinetic model of the reaction. Therefore, these possible reactant-catalyst interactions were ignored. To determine $k$ we used the following form of Eq. (3):

$$k_t = \frac{I_t}{I_f \left( I_f - I_t \right) t}, \tag{4}$$

which gives the value of $k_t$ measured for each time instance during the reaction progress. In the results, we present $k$ as the weighted mean of $k_t$ with weighted mean errors. A more detailed error analysis is available in Supplementary Note 3.

### Influence of ions and zwitterions on the reaction kinetics

In pure water, the energy transfer during diffusive collisions is often not enough to overcome repulsive interactions resulting from the

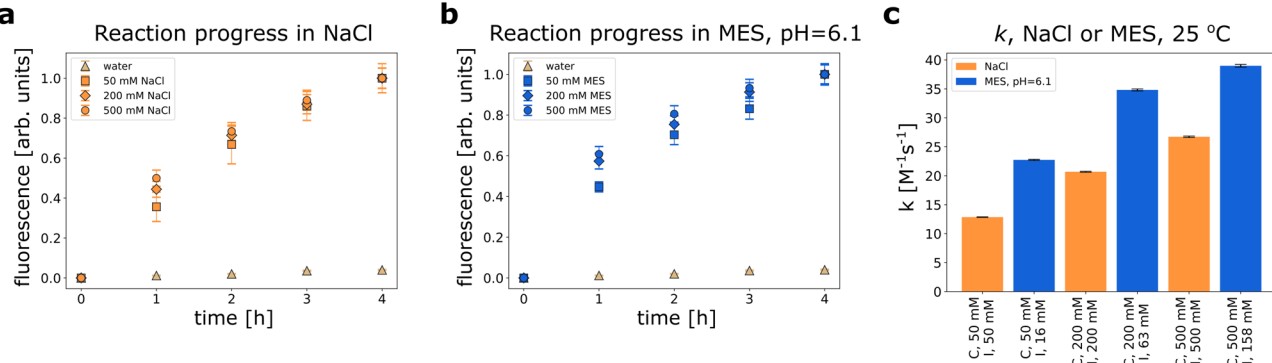

**Fig. 2 | Reaction in the presence of ions and zwitterions.** Reaction progress over time between 10 μM CoA and 10 μM CoA-M in the presence of **a** sodium chloride or **b** MES buffer pH 6.1 and **c** their reaction rate constants, $T = 25$ °C. Despite the lower ionic strength of the solution, zwitterions (MES) better accelerate the reaction between CoA and CoA-M than ions (sodium chloride). Error bars for **a** and **b** correspond to standard deviation, whereas for **c** correspond to weighted mean errors calculated as described in Supplementary Note 3. Source data are provided as a Source Data file.

negative charge on the highly charged molecules used in our study (four negative charges per one CoA molecule). To visualize this effect, we performed a reaction at 10 μM reactants solution (ratio 1:1) in pure MiliQ water, see Supplementary Fig. 3. The reaction between reactants is very slow ($k = 0.05$ M$^{-1}$ s$^{-1}$) and takes ~2 months. However, if we screen the reactants' charges with ions by adding sodium chloride, the process speeds up. Figure 2a depicts the influence of sodium chloride concentration changes on the reaction speed. For 50 mM NaCl solution, the reaction is ~250 times faster ($k = 12.9$ M$^{-1}$ s$^{-1}$) than in pure water. This change is due to the increase in ionic strength ($I$) and the screening of the negative charge of the reactants by ions. As a result, the Debye length and the effective negative charges of the reactants are reduced. In this situation, the same energy obtained by the collision of the reactants with the solvent molecules may be sufficient to form the product of the reaction. In Supplementary Fig. 2c, we show that change in ion concentration does not influence the photophysical properties of the fluorescent moiety.

Although zwitterions do not contribute to the ionic strength of a solution[8], we obtained a similar acceleration for the MES (zwitterion) buffer of 6.1 pH, see Fig. 2b. The $k$ for each MES buffer of a given concentration is ~2 times greater than the $k$ for the sodium chloride solutions of respective concentrations. These differences are shown in Fig. 2c. For example, for 50 mM MES buffer $k = 22.74$ M$^{-1}$ s$^{-1}$, $I = 16$ mM, and for the 50 mM NaCl solution $k = 12.86$ M$^{-1}$ s$^{-1}$, $I = 50$ mM. Zwitterions have no net charge, but they can associate with reactants reducing repulsive properties between them and, as a result, accelerating product formation. The values of $I$ for MES are different from 0 due to the presence of NaOH in the buffer solution. The effect of pH on the reaction is shown in Supplementary Note 6.

**Influence of net-charged molecules on the reaction kinetics**
The reaction between CoA and CoA-M can be accelerated by more complex molecules that possess a net charge, e.g., oligomers, enzymes, binding sites of proteins, etc. Such systems have the possibility of forming momentary non-covalent complexes that partially neutralize the reactants' charges and bring them together in close vicinity, enhancing reactants encounters. Therefore, in the second series of experiments, we introduced molecules with a positive net charge: arginine, nona-arginine (arg-9), or poly-L-lysine (135 mers) into the 50 mM MES buffer pH 6.1 solution. The concentration of both CoA and CoA-M was 10 μM, and the concentration of catalysts was set so that the number of positive charges in the solution was equal to the number of negative charges on the reactants. Thus, the concentration of arginine was 80 μM (10 μM times 8 negative charges), the concentration of nona-arginine was 8.9 μM, and the concentration of polylysine was ~0.6 μM. The reaction progress in the presence of

charged catalysts and the $k$ values are shown in Fig. 3a, b. The reaction speeds up with the increase in charge on the catalyst molecule (from $k = 4.32 \times 10$ M$^{-1}$ s$^{-1}$ for single charged arginine to $k = 1.24 \times 10^4$ M$^{-1}$ s$^{-1}$ for multiple charged polylysine).

By increasing the number of amino acids (i.e., arg, lys) in the peptide, the charge generated by single-molecule increases. The ionic strength varies with the square of the charge present on the molecule. An increase in catalyst concentration with multiple charges will cause a significant difference in the ionic strength of the system. For instance, a concentration increment of polylysine (having on average 135 charges) by 1 mM results in an 9.1 M higher ionic strength of the solution. However, due to the low concentration of polylysine (~0.6 μM) and arg-9 (8.9 μM), the ionic strength increases marginally compared to the ionic strength provided by the buffer. For instance, in the studied system, the addition of polylysine increases the ionic strength by 5 mM (to 21 mM), while the MES buffer alone contributes to 16 mM. Yet, polylysine ($k = 1.24 \times 10^4$ M$^{-1}$ s$^{-1}$) still accelerates the reaction by ~10$^5$ folds regarding the one in pure water ($k = 5 \times 10^{-2}$ M$^{-1}$ s$^{-1}$).

Due to the Coulomb interactions, oppositely charged particles associate with each other. Therefore, during the product formation, an intermediate of CoA-catalyst or CoA-M-catalyst occurs. This statement is supported by the change in fluorescence intensity growth versus concentration of fluorescent reactant after introducing nona-arginine or polylysine, see Supplementary Fig. 2b. Thus, in the case of net-positively charged molecules, the reaction occurs at their surfaces, as we previously hypothesized[6]. Figure 3b shows that the longer the chain is, the acceleration of the reaction is more significant. This acceleration is not achieved by an increase in charge concentration in the solution since we adjusted the concentration of catalysts to keep the charge concentration constant. The same amount of charges is located on fewer catalysts, and more reactants accumulate in their surroundings and stay there longer. Thus, the system consists of groups of reactants organized around the catalysts that can react with each other with greater probability due to their higher effective local concentration.

Although MES (zwitterion) and arginine (charged monomer) both have similar surface areas, $\hat{S}$, ($\hat{S}_{MES} = 3.92$ nm$^2$, $\hat{S}_{arg} = 3.72$ nm$^2$, both calculated with probe radius equal to 1.4 Å), and associate with reactant molecules, the reaction is more effectively accelerated by arginine. For instance, in the presence of 80 μM arginine added to the 50 mM MES buffer the reaction rate is 2 times higher than for a pure 50 mM MES ($k = 43.2$ M$^{-1}$ s$^{-1}$ vs $k = 22.7$ M$^{-1}$ s$^{-1}$, respectively) even though the added concentration of ions is 3 orders of magnitude lower than zwitterions provided by the buffer. Moreover, we increased further the zwitterions concentration by order of magnitude (using 500 mM MES pure buffer) and still obtained a lower reaction rate ($k = 39.0$ M$^{-1}$ s$^{-1}$) comparing to arginine. This simple example shows how crucial the

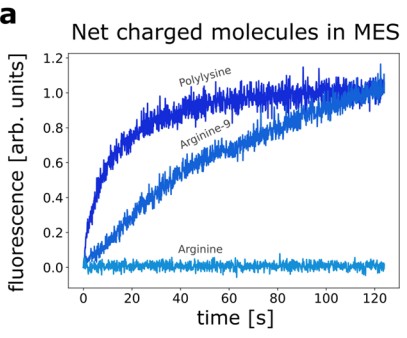
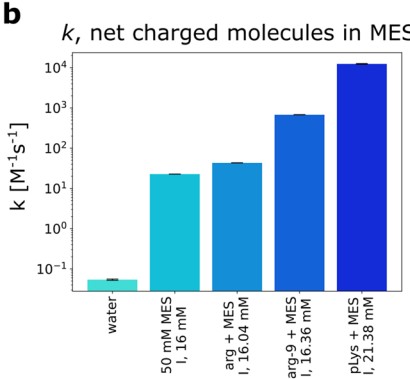
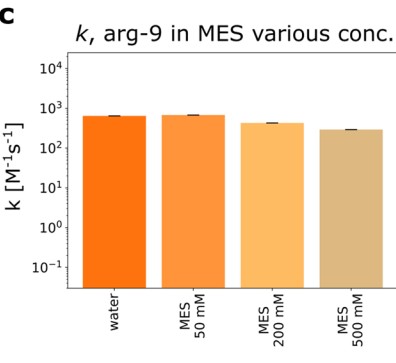

**Fig. 3 | Reaction in the presence of net-charged catalysts. a** CoA-M-CoA product formation in the presence of charged monomers, oligomers, and polymers, **b** their $k$ values. The concentration of catalysts is set, so the concentration of positive charges carried by them is equal to the concentration of negative charges on the

reactants; reaction in 50 mM MES pH 6.1, $T = 25$ °C. **c** $k$ values for the reaction in the constant concentration of nona-arginine and various concentrations of MES pH 6.1, $T = 25$ °C. Error bars correspond to weighted mean errors calculated as described in Supplementary Note 3. Source data are provided as a Source Data file.

arrangement of the charges within a molecule is for the catalytic effect, as well as its chemical structure and surrounding of the catalytically active part (MES as the zwitterion also possesses equivalent negative charge introduced by ethane sulfonic moiety).

Nevertheless, both increasing the concentration of zwitterions and molecules with positive net charge accelerate the reaction. However, if we keep a constant concentration of net-charged molecules and increase the concentration of zwitterions, the reaction slows down, which is depicted in Fig. 3c. It may seem counterintuitive when comparing this result with the previous ones, where the increase of any catalyst caused reaction acceleration. This deviation is due to the ion-dipole interaction occurring between zwitterions and net-charged molecules. If we increase the concentration of MES, net-charged catalysts associate with fewer reactant molecules, and hence product formation is slower.

### Influence of micelles on the reaction kinetics

Once we proved the acceleration with multi-charged compounds as polylysine, we decided to check whether cheap, commercially available, and well-known positively charged surfactants (also possessing multiple charges per micelle) as CTAC can affect the reactions similarly. In the case of CTAC, the reaction can be catalyzed either by free surfactants or more complex structures, micelles, which form above critical micelle concentration (CMC). The reaction should proceed at different speeds regarding the system composition, see Fig. 4a. Therefore, in the first step, we measured the influence of CTAC concentration on the CoA and CoA-M reaction above and below CMC. We performed measurements under identical experimental conditions as previously—in 50 mM MES pH 6.1 solution and 10 µM reactant concentration. Since experiments were conducted in an ionic solution, we calculated the CMC utilizing a volume-based thermodynamics model, see Supplementary Note 5[9]. The CMC of CTAC in water is 1.06 mM, and in 50 mM MES is 0.70 mM. The aggregation number for CTAC micelles is 81[10], so referring to micelle concentration, we mean concentration of surfactants above CMC divided by aggregation number. For example, the concentration of CTAC micelles in 1.35 mM CTAC solution in 50 mM MES is ~8 µM; (1.35–0.70 mM)/81.

We observed the highest $k$ value ($k = 1.92 \times 10^4$ M$^{-1}$s$^{-1}$) for the solution with a relatively low micelle concentration (~5 µM), see Fig. 4b, c. In this regime, the concentration of reactants is higher than the concentration of micelles, which suggests that more than one reactant molecule can attach to the micelle. Therefore, later interaction between CoA and CoA-M may result from the sliding at the micelle surface, as in the case of sliding at the polymer chain shown before[6]. Below CMC surfactants act similarly to arginine (single-

charged molecule). Thus, the reaction acceleration is not as significant as for micelles ($k = 26.14$ M$^{-1}$s$^{-1}$). Furthermore, we note that surfactants accumulate at the glass surface, decreasing their total concentration in the bulk solution. In Supplementary Note 7, we show that the reaction also proceeds at the glass surface.

When the concentration of micelles notably exceeds the concentration of reactants, the reaction slows down ($k = 31.19$ M$^{-1}$s$^{-1}$). This change results from the decrease of free reactants in the solution and a significant excess of micelles over them (~11 times). Most CoA and CoA-M adhere to the micelles, and they need to detach from them before finding other reactive molecules. This searching process between CoA and CoA-M is longer for greater concentrations of micelles.

Subsequently, we investigated the influence of MES molecules on the reaction progress in the presence of CTAC. We kept constant the concentration of CTAC at 1.35 mM and varied the concentration of MES. As seen from Fig. 4d increase in MES concentration slows down the product formation, and the maximum $k$ value is achieved in water ($k = 2.27 \times 10^4$ M$^{-1}$s$^{-1}$). As in the case of net-charged molecules, it results from ion-dipole interaction between surfactant and MES. This outcome emphasizes the ion-pairing process behind micelle and coenzyme interaction and suggests that interactions between the micelle catalysts and coenzymes are mainly driven by entropy.

Considering that the most significant acceleration with micelles that we achieved so far, was in aqueous solution with a low micelle/reactant concentration ratio (1:2), we extended measurements in water to even lower micelle/reactant ratios (i.e., 1:100). Results of these experiments are shown in Fig. 4e. We note the highest $k = 2.75 \times 10^5$ M$^{-1}$s$^{-1}$ for 0.2 µM CTAC micelle concentration in water (ratio 1:50). This non-monotonic behavior for high reactant/micelle ratio can be argued based on the dynamics of the reactants' events at the catalyst's surface. Therefore, we expect that an optimal catalyst concentration exists. We describe this issue in more detail in Supplementary Note 9.

Pleased with such a great acceleration for CTAC, we decided to examine whether other positively charged surfactants (benzethonium chloride - BTC and cetylpyridinium chloride - CPC) cause the same effect. As controls, we used uncharged (Brij L23) and negatively charged (Sodium dodecyl sulfate - SDS) micelles. We performed experiments in water since zwitterions combined with micelles give a less favorable effect. The concentration of micelles was kept constant at 20 µM, as well as the concentration of reactants. We determined the concentrations of micelles for remaining surfactants knowing their CMC values and aggregation numbers[11–13]. The outcome of the experiments are presented in Fig. 4f. Please note that the lower $k$ for CTAC ($k = 2.91 \times 10^3$ M$^{-1}$s$^{-1}$) compared to the previous measurements

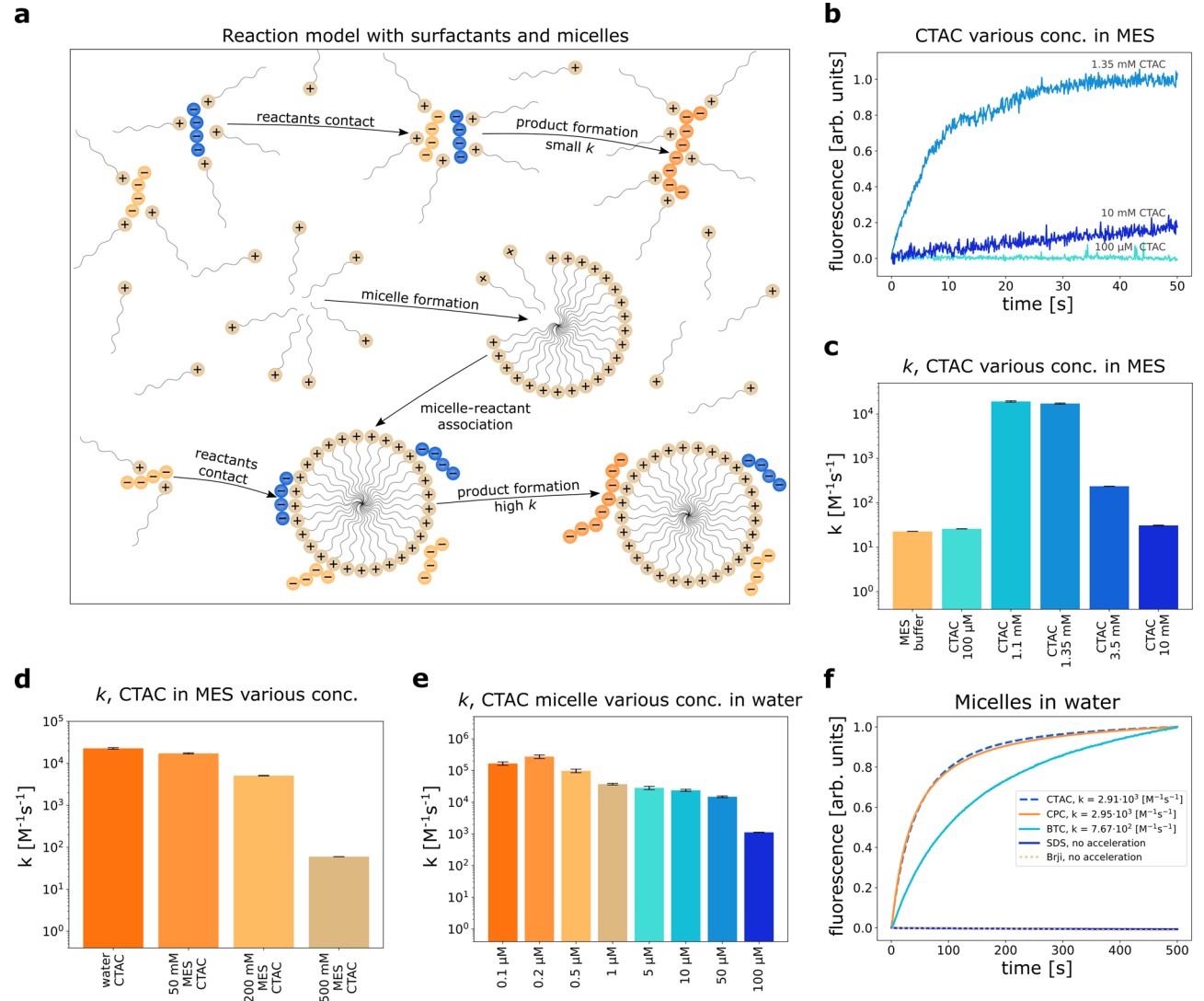

**Fig. 4 | Reaction in the presence of micelles. a** Reaction model with surfactants; product formation can be accelerated either by free surfactants or micelles with different speeds. **b** Reaction in 50 mM MES catalyzed with CTAC of different concentrations (above and below CMC, CMC = 0.7 mM), and **c** its $k$ values. **d** $k$ values for product formation in constact concentration of CTAC micelles (8 μM) and various concentrations of MES pH 6.1. **e** $k$ values for CoA/CoA-M reaction versus different CTAC micelle concentration in water (CMC = 1.06 mM) at constant concentration of each of reactants (10 μM). **f** Reaction progress in water with different micelles as catalysts (charged positively, negatively, or neutral) at constant micelle concentration (20 μM). Neutral (Brij) and negatively charged (SDS) micelles do not accelerate the reaction, which confirms the electrostatic nature of micelles-reactants interactions. All results in $T$ = 25 °C. Error bars for **c**–**e** correspond to weighted mean errors calculated as described in Supplementary Note 3. Source data are provided as a Source Data file.

in water arises from different micelle/reactants ratio (1:1). Results for BTC and CPC show that other positively charged micelles also generate great acceleration ($k_{CPC}$ = 2.95 × 10³ M⁻¹ s⁻¹). A slightly lower rate constant ($k_{BTC}$ = 7.67 × 10² M⁻¹ s⁻¹) for BTC comes from its structure. SDS and Brij do not increase the speed of product formation, which confirms the electrostatic nature of reactant-catalysts interactions. Therefore the observed effect is associated with the presence of charged species in the solution and not simply the occurrence of micelles.

### Second experimental system: DNA hybridization
To establish generality and validate whether the proposed acceleration method works for like-charged compounds in other systems, we introduced a second experimental model. We chose DNA hybridization because it is a well-known ion-sensitive process that occurs between two negatively charged reactants at neutral pH (Fig. 5a)[14]. This example also introduces broader generality since the product of CoA

with CoA-M reaction is formed via covalent bonds, whereas DNA duplex assembles via non-covalent interactions. To follow DNA hybridization, we monitored the time of reaching equilibrium between two single strands of oligonucleotides (thirteen nucleotides each). We investigated fluorescent-dye-labeled oligonucleotides by recording FRET transfer (Förster resonance energy transfer), the detailed description is in the Methods. Acquired data were analyzed with a second-order reaction model. We conducted experiments in 1 mM phosphate buffer (PB) at pH 7.4 and in water at constant oligonucleotides concentrations – 10 nM. In parallel, we monitored the hybridization for over 2 weeks in ultra-pure water. We revealed that no reaction occurs when counter-ions are absent. That is, the interaction is even slower than in the case of coenzymes, probably due to the larger overall charge of DNA strands. For more details, see Supplementary Fig. 6. However, after adding 1 mM of PB, the complete product is observed in ~70 h. Sodium cations screen negative charges on oligonucleotides enabling contact between them.

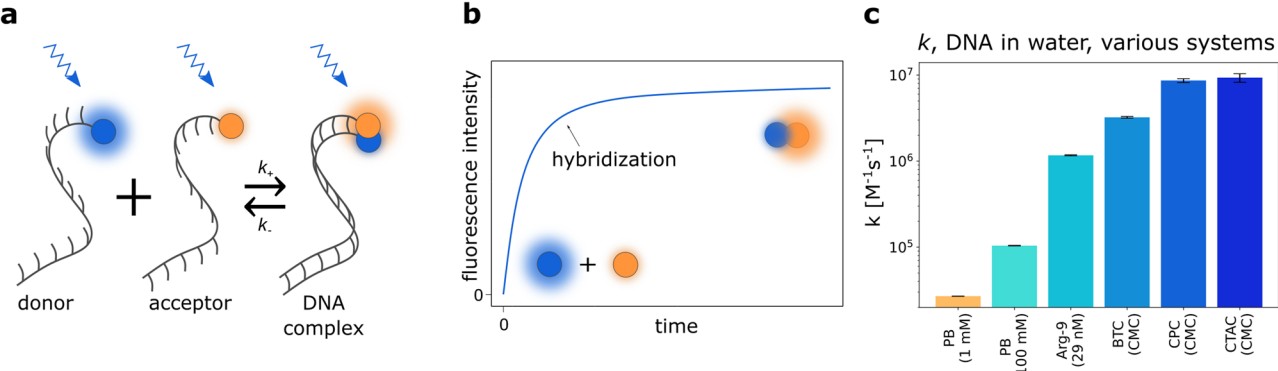

**Fig. 5 | Second reaction model – DNA hybridization. a** Model reaction between two DNA oligonucleotides (13 base pair each), one labeled with ATTO488 fluorophore (donor), and second with ATTO647N (acceptor). **b** Double strand formation was accompanied by an increase in fluoresence intensity of acceptor.

**c** Reaction rate constants of DNA double-strand formation in the presence of various catalysts in water with 0.002% TWEEN 20, $T = 25\,^\circ$C. Error bars for 5c correspond to weighted mean errors calculated as described in Supplementary Note 3. Source data are provided as a Source Data file.

Subsequent results are similar to the ones with coenzymes, which is depicted in Fig. 5c. Arginine-9 speeds up the reaction by 2 orders of magnitude and micelles by 3, regarding the one in 1 mM PB. Note that the results are slightly different in water and phosphate buffer solution (see Fig. 5c and Supplementary Fig. 6d). Again, the reaction did not take place in the presence of neutral and negatively charged micelles, confirming the electrostatic nature of catalysis (Supplementary Fig. 6f).

## Effect of Coulomb interactions on kinetics

The catalytic effect of molecules oppositely charged to reactants originates in enhanced particle transport. Therefore, it is independent of the details of specific reactant-reactant interactions. We support this claim experimentally, showing that the CoA reaction and DNA hybridization speedup can be explained with the same mechanism. We consider the electrostatic interactions between a single reactant particle and a catalyst particle. In ionic solution, in addition to the charge–charge interaction potential, it is necessary to take into account the electrical double layers. The presence of ions brings to the system Debye length $\lambda_D = \left(\frac{\varepsilon_0 \varepsilon k_B T}{e^2 \sum_i n_i}\right)^{1/2}$, which describes a screening distance for electrostatic interactions. For the 50 mM NaCl solution $\lambda_D \approx 1.4$ nm and for 50 mM MES at pH 6.1 $\lambda_D \approx 2.4$ nm. Further, we assume that the typical size of the catalyst is comparable to or larger than the Debye length. Therefore, it is reasonable to neglect the total charge of the catalyst as any probe charge placed in the vicinity of the catalyst does not sense it. Instead, we focus on the surface charge density $\chi$ present on the catalyst's surface. Using the Debye–Hückel approximation near a flat surface (see Supplementary Note 9, The influence of a large oppositely charged catalyst), we calculate the change in electrostatic energy that is experienced by a reactant, carrying charge $q_r$ between two positions: at the charged surface of the catalyst and in the bulk

$$\kappa = \frac{q_r \chi \lambda_D}{2\pi \varepsilon_0 \varepsilon k_B T},\tag{5}$$

which we scale with the energy of thermal fluctuations. Here $\varepsilon_0$ is the electric permittivity of vacuum, and $\varepsilon$ is the relative permittivity of water. Next, we solve Smoluchowski's equation[15, 16] to calculate the total flux of particles from the bulk to the flat surface where a chemical reaction occurs (see Supplementary Note 9, The influence of a large oppositely charged catalyst). This flux is often associated with reaction rate $k^s$ that is limited by the reactant transport. Although the exact formula for the whole spectrum of $\kappa$ is not known explicitly, we can

analyze it in the limit of $\kappa \ll -1$, where we find that

$$k^S = -k_0^S \kappa \tag{6}$$

with $k_0^S$ being a proportionality constant. Indeed, among our catalytic particles (in 50 mM MES) for CoA-Arg pair $\kappa \approx -13.8$ and for CoA-Arg9 pair $\kappa \approx -208$ and for other pairs, $\kappa$ has much more negative values. We acknowledge that the proportionality constant $k_0^S$ is theoretically hard to determine because of the parameters' uncertainty. Fortunately, the knowledge of $k_0^S$ is not necessary. The catalyst effectively increases reactant-reactant encounters, consequently increasing the reaction rate. The reaction rate, in the end, depends on reactant-reactant chemistry and is measured and not calculated from the first principles. Under this assumption, we have

$$k = -k_0 \kappa,\tag{7}$$

where $k_0$ can be found from experiments with different catalysts and a given reactant. We confirm this hypothesis by plotting $k/k_0$ as a function of $\kappa$ for CoA and DNA (Fig. 6). For each reactant, $k_0$ has to be determined by fitting a function of the form $k = -k_0 \kappa$ to experimentally measured $k$ with $\kappa$ calculated using parameters as in the experiment. We observe a linear dependence to an excellent approximation.

The existence of maximum reaction rate as a function of catalyst concentration supports the thesis that the catalyst acts primarily to increase the local concentration of the reactant in the vicinity of its surface (see Supplementary Note 9, The influence of the catalyst concentration). Asymptotically, for low catalyst concentrations, each catalyst particle generates sufficiently high local concentrations of the reactant at its surface that the flux of reactants from the bulk to the catalyst's surface limits the reaction rate. Increasing the catalyst concentration in the regime of very low concentrations results in increasing the total reaction rate. For large catalyst concentrations, there are many catalyst particles compared to reactants, and it rarely happens that two reactants reach the same catalyst particle. Therefore the action of the catalyst is hindered. In between, there is an interplay between those two effects. Surprisingly, we observe a weak dependence of the reaction rate on the catalyst concentration in this regime (Fig. 4e). The 250-fold change in catalyst concentration changes the reaction rate approximately 10 times, which further supports the hypothesis that the biggest changes in the reaction rate can be explained with $\kappa$ (Fig. 6). Weak dependence of the catalytic effect as a function of the concentration of the catalyst over its wide range (except concentrations close to 0 and much larger than reactants concentration) is a surprising effect. It encourages further research

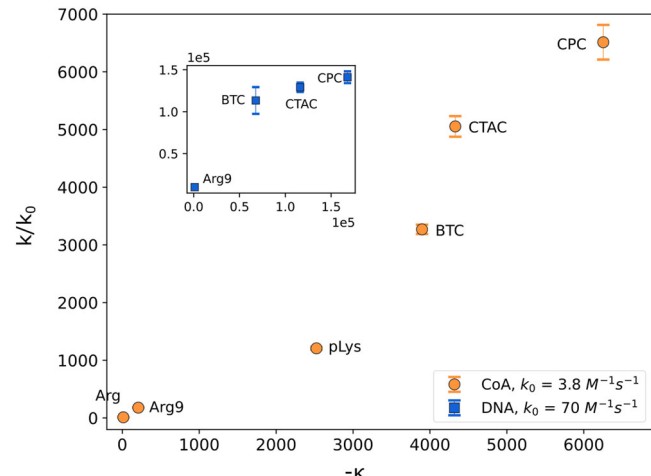

**Fig. 6 | Theoretical model.** Reaction rate increases as a function of $-\kappa$ for CoA and DNA. The values of $k_0$ are obtained from the fit of the linear function $k = -k_0\kappa$ to experimental data for each reactants. Error bars correspond to weighted mean errors calculated as described in Supplementary Note 3. Source data are provided as a Source Data file.

that focuses specifically on the reaction dynamics at the surface of the catalyst.

In summary, we have shown that the acceleration of the reaction rates between two like-charged (bio)molecules depends on the nature and form of the charge in the aqueous solution. By effectively screening the charge of the reactants, we speeded up the process of Coenzyme A molecules product formation. The reaction rate constant in relation to pure water increased ~$10^3$ folds by introducing ions or zwitterions (i.e., conventional screening). More importantly, we showed that by adding net charged molecules (monomers, oligomers, or polymers), we surpass this result by reaching a $10^3$–$10^5$-fold increase in reaction rate and even more than $10^6$-fold by adding positively-charged micelles. We explain this mechanism with an increase of local reactant concentration at the surface of positively charged particles due to the Coulombic attraction between reactants and catalysts that overwhelms repulsion between the reactants. When the same amount of charges was located on fewer catalysts (more charges per catalyst), as well as when the concentration of micelles was smaller than the concentration of reactants, the reaction proceeded in the fastest manner.

To demonstrate the general character of our methodology, we also studied DNA hybridization. Although the DNA double strands are associated through hydrogen bonds, and Coenzymes A product is formed through covalent bonds, the acceleration was found to be similar. Based on the obtained results, we proposed a theoretical model that can predict the acceleration of a particular reaction or interaction between like-charged molecules by oppositely charged catalysts. The model confirms that the catalytic effect is due to the electrostatic increase of reactant flux towards the surface of the catalyst.

Our results show that reaction (interaction) rates can be controlled within several orders of magnitude by tuning the magnitude and spatial distribution of the electric charge of the catalyst. This property will be highly useful for future research as a part of logical or sensing applications. For example, the rate acceleration could be used to detect the presence of specific ions in a solution or could be used to amplify signals from very dilute solutions.

## Methods
### Materials
We performed measurements utilizing two different reaction models with negatively charged reactants. The first one was the reaction

of Coenzyme A (CoA) (Sigma-Aldrich, USA) and bromo-N-methylmaleimide substituted CoA (CoA-M). We synthesized CoA-M and the reaction product (CoA-M-CoA) according to the "CoA-M and CoA-M-CoA synthesis" section. The reactants and the product were diluted to 500 µM in two series of aliquots in Milli-Q water and 50 mM MES buffer. Aliquots were stored at −20 °C. The second model was the reaction of double-strand complex formation between 13 base pair (bp) complementary DNA oligonucleotides (5′ ATC GTG TAG GCA T 3′, IBA GmbH, Germany). Strands were labeled with two dyes at the same end after hybridization - ATTO488 and ATTO647N. Oligonucleotides were stored at −20 °C in standard Tris-EDTA buffer as 100 µM stocks. The designed model prevents the formation of secondary structures.

For CoA/CoA-M reaction as the medium, we used Milli-Q ultrapure water (conductivity = 0.07 µS/cm) or 50 mM MES buffer, 2-(N-morpholino)ethanesulfonic acid (Sigma, USA), dissolved in water. The pH of buffer was adjusted by sodium hydroxide and kept constant at pH 6.1. In the case of the oligonucleotides, we used Milli-Q ultrapure water as the medium or 1 mM phosphate buffer (PB). Since the strands' concentrations were kept at a nanomolar scale, we added 0.002% Tween 20 to prevent their accumulation on the glass and air surfaces[17]. The temperature during all measurements was 25 °C.

As catalysts for the reaction, we used positively charged monomer - arginine (Anaspec, Belgium); positively charged oligomer - arginine-9 (Anaspec, Belgium); positively charged polymer - poly-L-lysine hydrochloride (Sigma-Aldrich, USA); positively charged surfactants: cetrimonium chloride (TCI, Belgium), cetylpyridinium chloride (Sigma-Aldrich, USA), benzethonium chloride (Sigma-Aldrich, USA); negatively charged surfactant - sodium dodecyl sulfate (GmbH, Germany), and neutral surfactant - Brij L23 (Sigma-Aldrich, USA). All catalysts were diluted in water. The surfactants were stored at 5 °C, while arginine, arginine-9, and polylysine were stored in −20 °C and unfroze before use.

### CoA-M and CoA-M-CoA synthesis
We synthesized CoA-M by the following steps. We dissolved 3,4-dibromo-1-methyl-1H-pyrrole-2,5-dione (0.7 mmol) in 2 ml of DMF and CoA (0.013 mmol) in 1 ml of DMF/water solution (volume ratio 9:1). Subsequently, we added dropwise CoA mixture into the reaction solution and left it overnight stirring at room temperature under argon. We removed the solvent in vacuo and purified the crude product extracting it with Et₂O/water. The product was acquired by freeze-drying of the aqueous phase.

To obtain CoA-M-CoA complex we dissolved 3,4-dibromo-1-methyl-1H-pyrrole-2,5-dione (0.006 mmol) in 1 mL of water. In the next step, we added CoA (0.013 mmol) into the reaction solution and stirred the mixture overnight at room temperature under argon. The product was acquired by freeze-drying.

### Microscope setup
Measurements were performed on Nikon C1 inverted confocal microscope equipped with the PicoQuant LSM module. The system was supported by the PicoHarp 300 Time-Correlated Single-Photon Counting setup (TCSPC). We set the parameters of the focal volume using a Nikon PlanApo water immersion lens 60× (NA = 1.2).

The fluorescent reactant can absorb light at a regime of 320–500 nm wavelength, see Supplementary Fig. 1. Thus, to excite the samples, we used a pulsed diode laser (485 nm) with a pulse frequency of 40 MHz (PicoQuant GmbH, Germany). The laser power was optimized to prevent photobleaching of the reactants. Hence, it was set on average at 100 ± 5 µW (power meter PM 100, Thorlabs, measured at a position in front of the light entering the objective). The fluorescence signal was collected by a single-photon avalanche photodiode (PerkinElmer Optoelectronics, Canada) after transmission of the photon signal through the 488 long-pass filter. To prevent noise detection and control temperature, we performed

measurements in the shaded climate chamber at $25 \pm 0.5\,°C$ (OkoLab, Italy). The samples were loaded into the glass container (ibidi GmbH, Germany) with the electromagnetic stirrer (4 mm diameter), and the focal volume was positioned 10 µm from the glass bottom.

We estimated the size of the focal volume by system calibration using Rhodamine 110 (Sigma-Aldrich) and performing the fluorescence spectroscopy measurement. Further modification of parameters was controlled with the PicoQuant Sepia II laser controller and the SymphoTime 64 software. We analyzed acquired data by the self-written Python script.

**FRET**

To perform FERT analysis using our microscope setup we acquired the data using pulsed interval excitation and Time-Correlated Single Photon Counting (TCSPC). TCSPC system records two intervals of time: the first, between start of the measurement and acquisition of the photon, and the second between laser pulses. The recorded photon data in the function of the time after the laser pulse can be presented as a histogram. The histogram represents the probability of recording photons after excitation, which can be understood as a fluorescence lifetime decay function. Using two intervaled excitation lasers enables to distinguish the emitted photon by a specific time gate (given laser excitation) and the color of an emission channel. For a sample with only the ATTO488-labeled strand, after the blue pulse, the photons are observed only in the blue channel, whereas after the red pulse only Instrument Response Function (IRF) is recorded. When a DNA duplex is formed, the donor (ATTO488) energy is absorbed after a blue laser pulse is transferred to the acceptor (ATTO647N), and fluorescence is becoming observed in the red channel. To record the kinetics of the DNA hybridization photon data were analyzed in the red channel (645 nm long pass filter, Chroma) after 485 nm laser pulse.

**Stopped-flow setup**

We also investigated the kinetics of the reaction by using stopped-flow fluorometer SX20 (Applied Photophysics Ltd., UK) and numerical analysis of recorded transients[18–20]. Stopped-flow experiments consisted of mixing two solutions at a 1:1 volume ratio in the mixing cell of the apparatus and registration of temporal changes of the fluorescence collected from the cell.

Both reactants were dissolved in an aqueous solution with catalyst as a co-solvent of even concentration in the injectors. Thus, the mixing did not result in catalyst dilution. However, in the case of co-solvents forming micelles, it is not clear if shearing forces arising in the mixing device of the stopped-flow apparatus do not introduce temporal changes in the concentration of micelles.

Samples were excited with a light-emitting diode (370 nm wavelength). The emission was collected at 90° to the excitation beam. A 550 nm long-pass emission filter was used (Schott OG550). The excitation and emission pathways were 5 and 1 mm, respectively.

## Data availability

The processed data are provided in Source Data file. Moreover, all the values of obtained $k$ constants with errors are summarized in Supplementary Tables 2 and 3. Source data are provided with this paper.

## Code availability

The code for this study (python script for $k$ constant fitting) is available here: https://github.com/AdamKowalskii/Effective-screening-of-Coulomb-repulsions.

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

## Acknowledgements

This work was supported by the National Science Center, Poland, within the grant Preludium Bis 2020/39/O/ST4/00877 to A.K. and R.H. W.T.S.H. was supported by the Simons Collaboration on the Origins of Life (SCOL; award 477123) and a Spinoza Grant of the Netherlands Organisation for Scientific Research (NWO). The authors thank Stanislaw Paczesny for helping us in setting up the measurement system. PJZ was supported by

European Union's Horizon 2020 research and innovation program under the Marie Skłodowska-Curie grant agreement No. 847413 and was a part of an international co-financed project founded from the program of the Minister of Science and Higher Education entitled "PMW" in the years 2020–2024; agreement no. 5005/H2020-MSCACOFUND/2019/2.

## Author contributions

A.K., K.B., and G.B. carried out the most experiments, conceptualization, methodology, validation, data analysis and wrote the original draft. P.J.Ż. developed a theoretical model and analyzed data. M.C. carried out stopped-flow experiments, and analyzed data. V.S. was responsible for reactants synthesis, validation, supervision, review, and editing. J.M.A. was responsible for helping carry out stopped-flow measurements, data analysis, supervision, review, and editing. W.T.S.H. was responsible for the supervision, formal analysis, and validation. R.H. was responsible for administration, conceptualization, supervision, formal analysis, funding acquisition, methodology, validation, review, and editing. All authors wrote the paper. All authors provided comments and agreed with the final form of the manuscript.

## Competing interests

The authors declare no competing interests.
