## [Peer Review File · Nature Communications]

REVIEWER COMMENTS

Reviewer #1 (Remarks to the Author):

The authors study a reaction CoA and CoA-M are report a change in rate constant depending in the presence of electrolytes, zwitter ions, charged monomers, oligomers, and polymers. The authors show that in the presence of macro-ions can rapidly accelerate the reaction. The authors hypothesize that the Coulomb interactions and electrostatic screening is responsible for the rapid increase in the reaction rate.

Overall, the experimental results are certainly interesting and could be useful for the broader chemistry community. The acceleration in rate constant is striking and I do think the authors have discovered a systematic trend in their system.

However, the authors haven't demonstrated the effected universally enough to warrant a publication in Nature Communications. Alternatively, the authors haven't explained / provided a predictive model that can capture some of these effects and explain the boost in reaction rate constant. Therefore, I am unable to recommend the publication in Nature Communication.

I would kindly request authors to focus on a predictive model "k" that is able to capture the experimentally observed effects quantitatively, or at least qualitatively. Or extend their analysis to 2-3 different experimental reaction systems to establish generality.

Reviewer #2 (Remarks to the Author):

I think the results of this manuscript are interesting and worth to be considered for publication. What is less clear to me is the mechanism and the impact of these results in a wider context. For the mechanism, I wonder if the reactants are screened, are they still be able to react? Would this screening prevent the formation of new bonding during the reaction, if for example there is complete screening of substrates? If not, how and why? Also, it is not quite clear how the micelles work, do they act as surfaces whereby reactants are adsorbed onto it, or else?

The explanation of the results in Figure 4c is also not clear, it seems there is an optimal concentration of CTAC, too high or too low would be detrimental, the authors attempt to explain why but I do not fully understand that.

As for the impact, it is not clear how such findings could be used to advance the field or related fields in Chemistry.

Response to Reviewers

Response to Reviewer: 1

Reviewer 1: The authors study a reaction CoA and CoA-M are report a change in rate constant depending in the presence of electrolytes, zwitter ions, charged monomers, oligomers, and polymers. The authors show that in the presence of macro-ions can rapidly accelerate the reaction. The authors hypothesize that the Coulomb interactions and electrostatic screening is responsible for the rapid increase in the reaction rate.

Overall, the experimental results are certainly interesting and could be useful for the broader chemistry community. The acceleration in rate constant is striking and I do think the authors have discovered a systematic trend in their system.

Author reply: We thank Reviewer 1 for his/her favorable comment.

Reviewer 1: However, the authors haven't demonstrated the effected universally enough to warrant a publication in Nature Communications.

Alternatively, the authors haven't explained / provided a predictive model that can capture some of these effects and explain the boost in reaction rate constant. Therefore, I am unable to recommend the publication in Nature Communication.

I would kindly request authors to focus on a predictive model "k" that is able to capture the experimentally observed effects quantitatively, or at least qualitatively. Or extend their analysis to 2-3 different experimental reaction systems to establish generality.

Author reply: We have followed this comment and realized both of the optional requests stated by Reviewer 2. As a result we:

- added a second experimental reaction system based on DNA hybridization. We accelerated the DNA duplex formation by orders of magnitude using exactly the same positively charged compounds as in the original approach (Manuscript pages 10-11, lines 289-316, Figure 5 and Supplementary Information page S6). This model widens our analysis as the DNA double-strand associates through hydrogen bonds, whereas the Coenzymes A product forms through covalent bonds.

- amended the manuscript with the theoretical model predicting and explaining quantitatively observed effects (Manuscript pages 11-13, lines 317-382, Figure 6 and Supplementary Information pages S7-S10). Moreover, the model/theory not only works for the original findings but also predicts effects shown on a second experimental system.

Taking into consideration both of the above points, we are fully convinced that we have established the required generality regarding our discovery and that the model can predict the speeding up of any particular reaction between like-charged reactants by oppositely charged accelerants in aqueous solutions.

Response to Reviewer: 2

Reviewer 2: I think the results of this manuscript are interesting and worth to be considered for publication. What is less clear to me is the mechanism and the impact of these results in a wider context.

Author reply: We thank Reviewer 2 for his/her favorable comment. We amended manuscript to clarify Reviewer's concerns. We extended the manuscript with theoretical bases that were followed with proper model and simulations (Manuscript pages 11-13, lines 317-382, Figure 6 and Supplementary Information pages S7-S10).

Reviewer 2: For the mechanism, I wonder if the reactants are screened, are they still be able to react? Would this screening prevent the formation of new bonding during the reaction, if for example there is complete screening of substrates? If not, how and why?

Author reply: To answer these questions directly: if we treated reactants as charged points in space, the reaction would not be possible, as the screened Coulomb potential would be identical after iteration by the same solid angles. If we reduced the potential energy of reactants (including all interactions) to 0, creating a product would have no energy benefit. In the case of actual molecules, not points, there would be the presence of spatial configurations enabling reactants to come in close proximity and allowing them to still undergo reactions. In other words: any reaction between like-charged reactants will be accelerated by oppositely charged catalysts in water solutions. Reactants have to be screened in order to react effectively. Indeed, our results showed that in lack of screening (i.e., water or low ionic strength conditions), the reaction occurs significantly slower than in high ionic strength conditions and orders of magnitude slower than in the presence of oppositely charged accelerants (e.g., Figure 2c and Figure 3b). Moreover, new experimental data (Figure 4e) show that in an aqueous solution, the high surplus of screening molecules (micelles) to reactants (i.e., 10:1) does not prevent new bond formation as we observed the product of a covalent reaction between CoA and CoA-M.

Reviewer 2: Also, it is not quite clear how the micelles work, do they act as surfaces whereby reactants are adsorbed onto it, or else?

Author reply: In the example of polylysine, we expect reactants to localize on the surface of the polymer and then react. In the case of micelles, we also expect the same behavior. One might suggest that reaction undergoes within the volume of micelles, and an increase of kinetics is observed due to the nano compartmentalization. However, the partitioning of charged molecules inside the hydrophobic interior of micelles is unlikely and experiments showed that the micelles formed by neutrally charged surfactant, i.e., Brij L23 generate no increase in kinetics.

Reviewer 2: The explanation of the results in Figure 4c is also not clear, it seems there is an optimal concentration of CTAC, too high or too low would be detrimental, the authors attempt to explain why but I do not fully understand that.

Author reply: As Reviewer 2 noticed, indeed, there is an optimal concentration ratio between reactants and catalysts. When the concentration of micelles significantly exceeds the concentration of reactants, the reactants hardly meet at the same micelle. We added the following sentences to the Manuscript (page 9, line 244-250) to make it clear: *“When the concentration of micelles notably exceeds the concentration of reactants, the reaction slows down ($k = 68.01 \text{ M}^{-1}\text{s}^{-1}$). This change results from the decrease of free reactants in the solution and significant excess of micelles over them (~11 times). Most CoA and CoA-M adhere to the micelles, and they need to detach from them before finding other reactive molecules. This searching process between CoA and CoA-M is longer for greater concentrations of micelles.”*

However, when the concentration of reactants exceeds the concentration of micelles, the results were less clear. To answer this question, we conducted additional experiments on Coenzymes A product formation in the presence of lower CTAC micelle concentrations in water. We showed new results in Figure 4e page 8 and added the following sentences to the Manuscript (page 9, line 260-269): *“Considering that the most significant acceleration with micelles that we achieved so far, was in aqueous solution with a low micelle/reactant concentration ratio (1:2), we extended measurements in water to even lower micelle/reactant ratios (i.e., 1:100). Results of these experiments are shown in Figure 4e. We note the highest $k = 2.75 \cdot 10^5 \text{ M}^{-1}\text{s}^{-1}$ for 0.2 μM CTAC micelle concentration in water (ratio 1:50). This non-monotonic behavior for high reactant/micelle ratio can be argued based on the dynamics of the reactants' events at the catalyst's surface. Therefore, we expect that an optimal catalyst concentration exists. We describe this issue in more details in Supplementary Information Section 9.4.”*

We note the optimal micelle/reactants concentration ratio as 1:50 and theoretically explain its dependence in the Supporting Information page S9, section 9.4. The influence of the catalyst concentration. In addition, we added in the main manuscript text following lines (page 12, lines 363-382): *The existence of maximum reaction rate as a function of catalyst concentration supports the thesis that the catalyst acts primarily to increase the local concentration of the reactant in the vicinity of its surface (see Supplementary Information Section 9.4). Asymptotically, for low catalyst concentrations, each catalyst particle generates sufficiently high local concentrations of reactant at its surface that the flux of reactants from the bulk to the catalyst's surface limits the reaction rate. Increasing the catalyst concentration in the regime of very low concentrations, results in increasing the total reaction rate. For large catalyst concentrations, there are many catalysts particles compared to reactants, and it rarely happens that two reactants reach the same catalyst particle. Therefore the action of the catalyst is hindered. In between, there is an interplay between those two effects. Surprisingly, we observe a weak dependence of the reaction rate on the catalyst concentration in this regime (Figure 4e). The 250-fold change in catalyst concentration changes the reaction rate approximately 10 times, which further supports the hypothesis that biggest changes in the reaction rate can be explained with κ (Figure 6). Weak dependence of the catalytic effect as a function of the concentration of the catalyst over its wide range (except concentrations close to 0 and much larger than substrate concentrations) is a surprising effect. It encourages further research that focuses specifically on the reaction dynamics at the surface of the catalyst.*

Reviewer 2: As for the impact, it is not clear how such findings could be used to advance the field or related fields in Chemistry.

Author reply: The new applied model can be utilized to predict the speeding up of any particular reaction with likely charged reactants by oppositely charged catalysts in aqueous solutions. This feature will be highly useful for future research as a part of logical or sensing applications. For example, the rate acceleration could be used to detect the presence of specific ions in a solution or could be used to amplify signals from very dilute solutions. Additionally, by applying a specific concentration of catalysts, one can control the reaction speed within orders of magnitude.

ORIGINAL REVIEWER COMMENTS

Reviewer #1 (Remarks to the Author):

The authors study a reaction CoA and CoA-M are report a change in rate constant depending in the presence of electrolytes, zwitter ions, charged monomers, oligomers, and polymers. The authors show that in the presence of macro-ions can rapidly accelerate the reaction. The authors hypothesize that the Coulomb interactions and electrostatic screening is responsible for the rapid increase in the reaction rate.

Overall, the experimental results are certainly interesting and could be useful for the broader chemistry community. The acceleration in rate constant is striking and I do think the authors have discovered a systematic trend in their system.

However, the authors haven't demonstrated the effected universally enough to warrant a publication in Nature Communications. Alternatively, the authors haven't explained / provided a predictive model that can capture some of these effects and explain the boost in reaction rate constant. Therefore, I am unable to recommend the publication in Nature Communication.

I would kindly request authors to focus on a predictive model "k" that is able to capture the experimentally observed effects quantitatively, or at least qualitatively. Or extend their analysis to 2-3 different experimental reaction systems to establish generality.

Reviewer #2 (Remarks to the Author):

I think the results of this manuscript are interesting and worth to be considered for publication. What is less clear to me is the mechanism and the impact of these results in a wider context. For the mechanism, I wonder if the reactants are screened, are they still be able to react? Would this screening prevent the formation of new bonding during the reaction, if for example there is complete screening of substrates? If not, how and why? Also, it is not quite clear how the micelles work, do they act as surfaces whereby reactants are adsorbed onto it, or else?

The explanation of the results in Figure 4c is also not clear, it seems there is an optimal concentration of CTAC, too high or too low would be detrimental, the authors attempt to explain why but I do not fully understand that.

As for the impact, it is not clear how such findings could be used to advance the field or related fields in Chemistry.

REVIEWERS' COMMENTS

Reviewer #1 (Remarks to the Author):

I thank the author for taking my comments seriously. The authors have made two important additions to the manuscript. First, they have shown that their discovered effects are also application to a new system, i.e., Fig. 5. Second, they have also quantified the increase in rate, i.e., Fig. 6.

With these changes, authors have addressed my concerns. I am happy to recommend the paper for publication Nature Communications.

The one minor comment I do have is for Fig. 6, where the font sizes are significantly larger compared to other figures.

Reviewer #2 (Remarks to the Author):

I have carefully read the answers of the authors to both my comments and those of the other referee, which are quite reasonable and detailed. I recommend the revised manuscript to be published in Nature Communications as the study is well carried out, clearly presented, shows interesting results and follows a rigorous methodology.

Response to Reviewers

Response to Reviewer: 1

Reviewer #1 (Remarks to the Author): I thank the author for taking my comments seriously. The authors have made two important additions to the manuscript. First, they have shown that their discovered effects are also application to a new system, i.e., Fig. 5. Second, they have also quantified the increase in rate, i.e., Fig. 6.

With these changes, authors have addressed my concerns. I am happy to recommend the paper for publication Nature Communications.

The one minor comment I do have is for Fig. 6, where the font sizes are significantly larger compared to other figures.

Author reply: We thank Reviewer 1 very much for his/her favorable comment.

We have followed this one minor comment and decreased the font sizes in Figure 6.

Response to Reviewer: 2

Reviewer #2 (Remarks to the Author): I have carefully read the answers of the authors to both my comments and those of the other referee, which are quite reasonable and detailed. I recommend the revised manuscript to be published in Nature Communications as the study is well carried out, clearly presented, shows interesting results and follows a rigorous methodology.

Author reply: We thank Reviewer 2 very much for his/her favorable comment.